# CD11b Deficiency Favors Cartilage Calcification via Increased Matrix Vesicles, Apoptosis, and Lysyl Oxidase Activity

**DOI:** 10.3390/ijms24119776

**Published:** 2023-06-05

**Authors:** Ilaria Bernabei, Uwe Hansen, Driss Ehirchiou, Jürgen Brinckmann, Veronique Chobaz, Nathalie Busso, Sonia Nasi

**Affiliations:** 1Service of Rheumatology, Department of Musculoskeletal Medicine, Lausanne University Hospital, 1011 Lausanne, Switzerland; 2Institute for Musculoskeletal Medicine, University Hospital of Münster, 48149 Münster, Germany; 3Department of Dermatology, University of Lübeck, 23562 Lübeck, Germany

**Keywords:** CD11b, integrin, calcium-containing crystals, transmission electron microscopy, chondrocyte mineralization, cartilage calcification, lysyl oxidase

## Abstract

Pathological cartilage calcification is a hallmark feature of osteoarthritis, a common degenerative joint disease, characterized by cartilage damage, progressively causing pain and loss of movement. The integrin subunit CD11b was shown to play a protective role against cartilage calcification in a mouse model of surgery-induced OA. Here, we investigated the possible mechanism by which CD11b deficiency could favor cartilage calcification by using naïve mice. First, we found by transmission electron microscopy (TEM) that CD11b KO cartilage from young mice presented early calcification spots compared with WT. CD11b KO cartilage from old mice showed progression of calcification areas. Mechanistically, we found more calcification-competent matrix vesicles and more apoptosis in both cartilage and chondrocytes isolated from CD11b-deficient mice. Additionally, the extracellular matrix from cartilage lacking the integrin was dysregulated with increased collagen fibrils with smaller diameters. Moreover, we revealed by TEM that CD11b KO cartilage had increased expression of lysyl oxidase (LOX), the enzyme that catalyzes matrix crosslinks. We confirmed this in murine primary CD11b KO chondrocytes, where *Lox* gene expression and crosslinking activity were increased. Overall, our results suggest that CD11b integrin regulates cartilage calcification through reduced MV release, apoptosis, LOX activity, and matrix crosslinking. As such, CD11b activation might be a key pathway for maintaining cartilage integrity.

## 1. Introduction

Pathological calcification (PC) of articular cartilage is a hallmark of osteoarthritis (OA), a degenerative disease of the joints that affects globally over 300 million people [1] leading to increasing pain and movement restriction, with no effective treatment presently. PC is found in up to 100% of OA patients undergoing total knee or hip replacement surgeries and is correlated with cartilage degradation [2,3]. It is caused by cartilage-residing chondrocytes that become hypertrophic and apoptotic and produce hydroxyapatite crystals that are released in the extracellular matrix (ECM) [4,5]. In particular, OA chondrocytes release extracellular membrane-covered particles of around 30–150 nm in diameter, called matrix vesicles (MVs) [6]. MVs contain a calcification machinery of enzymes and transporters that concentrate calcium and phosphate, leading to crystal precipitation [4,7]. Ca^2+^ ions are internalized in MVs through the Annexin V (ANX V) channel, while extracellular PP_i_ is hydrolyzed by alkaline phosphatase (ALP) into P_i_ and transported inside MVs by Na/P_i_ cotransporters (PIT-1, PIT-2). Another mechanism of crystal formation is apoptosis, during which apoptotic bodies of 1000–5000 nm [8] are released. These are similar to MVs as they possess the same calcification machinery [5,9]; up to 10% of OA chondrocytes undergo apoptosis [10].

A novel integrin studied in OA cartilage is CD11b/CD18 (also known as α_M_β_2_ or Mac-1 for macrophage antigen 1), a member of the β_2_ family of leukocyte adhesion receptors, mostly studied for its anti- and pro-inflammatory capabilities in autoimmune diseases [11]. In our previous study, we demonstrated how CD11b knock-out (KO) murine chondrocytes were more prone to calcify, in comparison with WT cells, under mineralizing conditions [12]. Furthermore, lack of CD11b leads to increased cartilage damage in the in vivo OA mouse model.

Integrins mediate communication between cells and ECM, leading to the activation of intracellular signals, such as chondrogenic differentiation [13,14], metabolism, and matrix assembly [15]. Integrins are important sensors of ECM stiffness, as demonstrated by several cancer studies where integrin complexes are stabilized on stiff substrates [16,17]. This interaction activates upregulation of lysyl oxidase (LOX) and further matrix rigidity.

LOX is an enzyme that catalyzes the crosslinking of collagen and elastin fibrils [18]. Specifically, LOX modifies lysin residues to spontaneously form crosslinks called hydroxylysinonorleucine (HLNL), dihydroxylysinonorleucine (DHLNL), and histidinohydroxymerodesmosine (HHMD) [19]. Intriguingly, LOX is upregulated in OA cartilage [20], where ECM stiffening occurs in the early stages of OA before its degradation [21]. Moreover, LOX overexpression is associated with increased vascular calcification [22,23].

In the present study, we wanted to investigate the possible mechanism by which CD11b deficiency could favor cartilage calcification, which could lead to OA progression [12].

## 2. Results

### 2.1. CD11b-Deficient Cartilage from Naïve Mice Is More Calcified Than WT Cartilage

We previously demonstrated that CD11b-deficient chondrocytes were more prone to calcify than WT chondrocytes. Furthermore, CD11b KO mice exhibited increased cartilage calcification and degradation in a surgery-induced OA model [12].

Here we found that knee articular cartilage from naïve young CD11b KO mice shows early signs of calcification compared with cartilage from WT age-matched mice. Indeed, electron micrographs of the cartilage displayed increased mineral deposits in the form of druses in CD11b KO cartilage (Figure 1a, asterisks). This increase revealed statistically significant upon quantification by ImageJ (Figure 1a, graph). Interestingly, we found exacerbation of this difference in WT and CD11b naïve old mice (Figure 1b).

### 2.2. CD11b Deficiency Favors Matrix Vesicles’ Release by Chondrocytes

Matrix vesicles are membrane-invested particles where crystals form. We, therefore, analyzed their presence in WT and CD11b KO naïve cartilage. By TEM analysis, we noticed an increased distribution of electron-dense extracellular vesicles in CD11b KO cartilage compared with WT cartilage, both pericellular, then dispersed in the ECM (Figure 2a, arrowheads). The dimensions of these vesicles were between 50 nm and 200 nm, which are in agreement with the size of MVs [8]. Moreover, numerous needle-shaped mineral crystals were found close to the MVs. We quantified MVs and found a significant increase in CD11b KO cartilage compared with WT (Figure 2a, graph). To confirm the augmentation of MV-containing calcium crystals in CD11b KO cartilage, we serially ultracentrifuged supernatants of WT and CD11b KO chondrocytes to isolate MVs, and we measured the calcium content. In line with TEM data, we found significantly higher calcium in the CD11b KO sample (Figure 2b).

MVs are known to express two surface proteins important for crystal formation, ALP and ANX V. By immuno-gold labeling, we found high expression of ALP in CD11b KO cartilage, mainly localizing in the correspondence of MVs (Figure 2c, bold arrows), while WT cartilage expressed very little ALP spread in the ECM (Figure 2c, arrowheads). The corresponding quantification highlighted this difference (Figure 2c, graph). Similarly, ANX V was largely more expressed in CD11b KO cartilage, both extracellularly (Figure 2d, bold arrows and arrowheads) and in chondrocytes (Figure 2d, thin arrows), as also confirmed by its quantification (Figure 2d, graphs).

### 2.3. CD11b Deficiency Favors Chondrocyte Apoptosis

Another known mechanism of OA chondrocytes’ calcification is apoptosis. By TEM analysis, we found more signs of apoptotic chondrocytes in cartilage from naïve CD11b KO mice than in cartilage from WT mice. Indeed, the electron micrograph in Figure 3a shows an area within the extracellular matrix with the remaining parts of a dead chondrocyte (asterisk). The area has the same shape and size as a chondrocyte but the cytoplasm with the organelles is degraded and visible as numerous amorphous structures. In addition, the degraded chromatin appears as more electron-dense structures in this area. To confirm these in vivo observations, we isolated chondrocytes from WT and CD11b KO mice and performed FACS analysis for ANX V staining and for 7-amino-actinomycin staining (7AAD). ANX V/7AAD double staining is a convenient way to discriminate early apoptosis from late apoptosis and necrosis. We found a higher percentage of early apoptotic chondrocytes (Figure 3b, quadrants Q1) in CD11b KO chondrocytes, as confirmed by quantification (WT: 15.4 ± 2.5 vs. CD11b KO: 22.3 ± 3.3, Figure 3b, graph).

### 2.4. CD11b-Deficient Cartilage Shows ECM Alterations

ECM is a known modulator of calcification [24,25]. We, therefore, studied ECM appearance in WT and CD11b KO naïve cartilage. In TEM images, we noticed unstructured and less-ordered matrices in CD11b KO cartilage compared with WT cartilage. Both pericellular (Figure 4a, arrowhead) and extracellular matrices (Figure 4b, arrowhead) were affected. We went into more detail and analyzed collagen fibrils dimension in these samples. We found in CD11b KO cartilage an increased number of collagen fibrils with smaller diameters, in comparison with WT (Figure 4c,d).

### 2.5. CD11b KO Cartilage Presents Increased LOX Expression and Activity

Increased LOX activity has been shown to induce vascular calcification [22,23]. Here, TEM analysis of CD11b KO healthy cartilage revealed increased LOX immuno-gold labeling, mainly localized in the ECM rather in chondrocytes (Figure 5a,b). In addition to an increased expression of LOX protein in CD11b KO chondrocytes we found increased expression of the *Lox* gene (Figure 5b) and of LOX activity (Figure 5c). Finally, we measured the end product of LOX activity, namely crosslinks. We found a slight increase of HLNL and HHMD crosslinks in CD11b KO matrices (HLNL: WT = 100.0% ± 6.8% CD11b KO: 110.8% ± 10.5%, HHMD: WT = 100.0% ± 11.0% CD11bKO = 107.0% ± 9.5%), while no difference was found in the DHLNL crosslink (DHLNL: WT = 100.0% ± 8.0% CD11b KO = 99.6% ± 4.1%) (Figure 5d).

### 2.6. CD11b Partially Acts via Sensing Matrix Crosslinks

Next, we wanted to understand whether CD11b is a sensor of matrix crosslinking and stiffness, as found for other integrins in cancer studies [16,17,26,27]. We performed a two-step experiment as described in the method section. Briefly, we cultured WT chondrocytes in the absence (NT) or presence of BAPN in order to obtain matrices with normal or decreased amounts of crosslinks, as described in other studies [28,29] (Appendix A). In a second step, WT and CD11b KO chondrocytes were replated on these pre-formed matrices and let to calcify. We found that WT cells calcified significantly less on lowly crosslinked matrices (BAPN) than on normally crosslinked matrices (NT) (Figure 6a, WT, alizarin red pictures and graph). We observed a similar scenario in CD11b KO chondrocytes but to a lower and non-statistically significant extent. By qRT-PCR analysis, we confirmed that WT chondrocytes cultured on BAPN-treated matrices expressed significantly fewer calcification genes (*Anx5*, *Pit1*, *Pit2*), while the same genes were not modulated in CD11b KO cells cultured in the same conditions (Figure 6b, left graphs). Similarly, the *Lox* gene was downregulated in WT but not in CD11b KO chondrocytes cultured on less crosslinked matrices (BAPN-treated) (Figure 6b, right graphs).

## 3. Discussion

We previously reported that the CD11b integrin subunit had a protective role against cartilage pathological calcification in osteoarthritis [12]. In particular, CD11b KO mice were more affected than WT in the OA-induced meniscectomy model.

As CD11b was never described in cartilage, we showed CD11b is expressed by chondrocytes, although by only a subset of cells [12]. Here, we confirmed these findings by performing a meta-analysis of available GEO open-access databases, where a CD11b-encoding gene was expressed in human and murine chondrocytes and cartilage (Appendix A).

In this manuscript, we demonstrated that CD11b KO mice exhibited early cartilage calcification, as revealed by electron microscopy of healthy 13-week-old mice. Indeed, we found calcification spots resembling the previously published aspect of crystals by TEM [30]. This calcification was then exacerbated in 77-week-old CD11b KO mice.

Moreover, we found that calcification could be accounted for by at least three underlying mechanisms (Figure 7). It is currently well accepted that a crucial step in calcium crystal deposition is the release of MVs of 30–150 nm [4]. We discovered that CD11b KO chondrocytes release more calcifying vesicles in cartilage in vivo and in chondrocytes in vitro. Calcium-containing crystals form in MVs due to the precipitation of P_i_ and Ca^2+^ ions. P_i_ is generated from PP_i_ through ALP and transported inside MVs by PIT-1 and PIT-2. Additionally, Ca^2+^ is internalized through the ANX V channel [4,7]. Here, a higher number of MVs was accompanied by increased ALP and ANX V staining in CD11b KO cartilage (Figure 7, green). However, we did not investigate whether other Ca^2+^ channels than ANX V were involved in chondrocyte calcification. These could encompass the TRPV1 channel, shown to contribute to osteoblast mineralization [31], or the TRPV4 channel involved in chondrocyte calcification [32].

An alternative mechanism leading to crystal formation is chondrocyte apoptosis, a process of programmed cell death that releases apoptotic bodies and P_i_ and Ca^2+^ in the ECM [9,33,34]. We observed an increased number of apoptotic chondrocytes in CD11b KO cartilage, both by TEM in vivo and FACS analysis in vitro (Figure 7, blue). Intriguingly, CD11b-CD18 was found capable of both delaying and enhancing apoptosis in neutrophils, through inhibition or stimulation of the Akt pathway [35].

During cartilage development, collagen fibrillogenesis is a crucial step to provide strength and compressibility to the ECM [36], and integrins have been found to play a central role in collagen assembly [37]. Our results revealed disorganized pericellular and extracellular cartilage matrices in CD11b KO mice. This dysregulation consisted of more abundant collagen fibrils with smaller diameters, appearing sparser throughout the matrix (Figure 7, collagen fibrils in grey). Interestingly, the lack of α2β1 integrin in tendons brought a higher number of collagen fibrils with reduced diameter [38]. In line with our results, mice with chondrocyte-specific deletion of the integrin subunit β1 displayed growth plate abnormalities with errors in adhesion, proliferation, shape, and actin organization of chondrocytes [39]. In the same mice, cartilage exhibited a sparse, disorganized collagen network, associated with erosion.

We further investigated the modulation of LOX, the crosslinking enzyme required for the final maturation of collagen and elastin fibrils in the ECM. In our analyses, LOX expression was upregulated both at the gene and the protein levels in CD11b KO chondrocytes and cartilage, respectively. Accordingly, LOX activity as well as the cross-link HLNL were also slightly increased in CD11b KO chondrocytes (Figure 7, red). LOX has been previously associated with OA, where it was found overexpressed in a surgery-induced murine model and in damaged human OA cartilage [20]. In another work, LOX expression was upregulated in osteoblast-like VSMCs, and its inhibition reduced the transdifferentiation and calcification of these cells [22]. 

Integrins are sensors of ECM stiffening, as demonstrated in several diseases where matrix disorganization plays a pathogenic role. In cancer studies, tumor cells are more proliferative and invading on stiff microenvironments through integrins sensing of ECM mechanical forces [16,17,26,27]. For instance, matrix stiffness has upregulated LOXL2 through the integrin α5β1 signaling pathway in hepatocellular carcinoma [40].

As such, we questioned whether CD11b is capable of sensing the degree of matrix crosslinks leading to negative feedback on LOX expression, ultimately resulting in decreased calcification. Our results indicate that CD11b could indeed act in part via ECM sensing. While WT chondrocytes plated on lowly crosslinked matrices suppressed *Lox* expression and crystal production, CD11b KO were partially resistant to these effects.

In conclusion, we have revealed that a lack of CD11b in cartilage resulted in early-OA features, such as calcification together with apoptosis, dysregulated ECM, and LOX increase. As such, CD11b activation might be a key pathway for maintaining cartilage integrity.

## 4. Materials and Methods

### 4.1. Mice

C57BL/6 WT mice and CD11b KO mice (on C57BL/6 background, kindly provided by Prof. Britta Engelhardt, University of Bern) were used. Animals were kept in a temperature-controlled and ventilated rack with a 12:12 h light cycle and with access to water and food ad libitum.

### 4.2. Transmission Electron Microscopy (TEM) and Immuno-Gold Labeling

WT and CD11b KO female mice (n = 4 for each genotype) were sacrificed at 13 weeks old via CO_2_ inhalation. Intact right knees were dissected and fixed overnight at 4 °C in 2% (*v*/*v*) formaldehyde and 2.5% (*v*/*v*) glutaraldehyde in 100 mM phosphate buffer (PBS) with pH 7.4 for morphological TEM analysis. Left knees were fixed overnight at 4 °C 2% (*v*/*v*) formaldehyde and 0.25% (*v*/*v*) glutaraldehyde in 100 mM phosphate buffer (PBS) with pH 7.4 for immuno-gold electron microscopic analysis. From the day after, samples were stored in 100 mM PBS at 4 °C.

WT and CD11b KO female mice (n = 4 for each genotype) were sacrificed at 77 weeks old via CO_2_ inhalation. Intact right knees were dissected and fixed overnight as described above for morphological TEM analysis.

Afterward, cartilage was carefully dissected from the knees. Tissue samples used for morphological analysis were postfixed in 0.5% (*v*/*v*) osmium tetroxide and 1% (*w*/*v*) potassium hexacyanoferrate (III) in 0.1 M cacodylate buffer for 2 h at 4 °C, followed by washing with distilled water. After dehydration in an ascending ethanol series from 30 to 100%, specimens were incubated two times in propylene oxide each for 15 min and embedded in Epon using flat embedding molds. Tissue samples used for immuno-gold electron microscopic analysis were rinsed in PBS after fixation, dehydrated in ethanol up to 70%, and embedded in LR White embedding medium (London Resin Company, Berkshire, UK) using beem capsules. Ultrathin sections were cut with an ultramicrotome, collected on copper grids, and negatively stained with 2% uranyl acetate for 15 min. For immuno-gold electron microscopy, ultrathin sections were incubated with 100 mM glycine in PBS for 2 min, washed with PBS, and blocked with 2% (*w*/*v*) BSA and 1% normal goat serum in PBS. For immuno-labeling, grids were incubated for 1 h at room temperature on drops of primary antibodies (ALP (GTX100817) from GenTex (Irvine, CA, USA), 1:25; ANX V (PA5-78784) from Invitrogen (Waltham, MA, USA), 1:100; and LOX (E-19) from Santa Cruz Biotechnology (Dallas, TX, USA), 1:100 diluted in PBS containing 1% (*v*/*v*) BSA-c (Aurion, PD Wageningen, The Netherlands) and 0.025% (*v*/*v*) Tween 20. After washing with the same solution, ultrathin sections were incubated with secondary antibodies (diluted 1:20) conjugated to 18 nm gold particles. After washing with distilled water, ultrathin sections were negatively stained with 2% (*w*/*v*) uranyl acetate for 10 min. Electron micrographs were taken at 60 kV with a Phillips EM-410 electron microscope using imaging plates (Ditabis, Pforzheim, Germany). For quantification of the immuno-gold labeling, the number of gold particles was counted on several fields of EM images of WT and CD11b KO animals. The determinations of the areas of mineralization, matrix vesicles, immuno-gold labeled proteins, and fibril diameters were performed by using ImageJ 1.52q (National Institutes of Health, Bethesda, MD, USA) [41]. Means and standard deviation were determined by standard procedure.

### 4.3. Murine Articular Chondrocytes

Chondrocytes were isolated from WT and CD11b KO mice between 4 and 7 days old, as previously described [42], and amplified for 7 days in not-treated medium (NT = DMEM with 10% FBS). Cells were replated at 5 × 10^4^ cells/cm^2^ at passage 2 in NT for all experiments.

### 4.4. Two-Steps Experiment

In the first step, we cultured murine WT chondrocytes for 21 days in DMEM complete medium (10% FBS, 50 µg/mL ascorbic acid), in the presence or absence of the pan-LOX(L) inhibitor β-aminoproprionitrile (BAPN 500 μM, Sigma-Aldrich, Burlington, MA, USA) to form matrices with various degrees of crosslinks (Appendix A). Cells were washed three times with 4 °C PBS and killed by dry O/N freezing at −20 °C. Then, the addition of 0.5% deoxycholate for 10 min at 4 °C was used to remove the cellular debris, and the matrices were washed three times with PBS. In the second step, WT and CD11b KO chondrocytes were plated on the pre-formed matrices and cultured for 3 days in calcifying BGJb medium (10% FBS, 50 µg/mL ascorbic acid, 20 mM β-glycerophosphate). Calcification was assessed by Alizarin red staining as previously described [12] and qRT-PCR for the indicated genes (Table 1) was performed.

### 4.5. MVs Isolation and Calcium Content

Extracellular vesicles of 30–150 nm diameter (corresponding to matrix vesicles, MVs) were isolated from chondrocytes supernatants by serial ultracentrifugations at 4 °C [8] using Optima™ XPN-80 (Beckman Coulter, Brea, CA, USA). Briefly, WT and CD11b KO chondrocytes were cultured for 24 h in DMEM with 10% FBS. Supernatants were collected and centrifuged at 300× *g* for 10 min to remove dead cells and debris. Then, a second centrifugation at 2000× *g* for 10 min was performed to pellet apoptotic bodies (1000–5000 nm). A third centrifugation at 10,000× *g* for 30 min allowed us to pellet and remove larger microvesicles (>100–1000 nm). The final centrifugation at 100,000× *g* for 70 min was done to pellet matrix vesicles. Pelleted MVs were resuspended in PBS, and RIPA buffer was added to disrupt the phospholipidic layer in order to measure calcium content with QuantiChromTM Calcium Kit (BioAssay Systems, Hayward, CA, USA).

### 4.6. Fluorescence-Activated Cell Sorting (FACS)

Chondrocytes were detached from culture plates by use of non-enzymatic cell dissociation buffer (5 mM EDTA in PBS). Cells were then suspended in FACS buffer (5 mM EDTA, 3% FBS in PBS) and labeled with AnnexinV-PE (Apoptosis Detection Kit, eBioscience^TM^, Waltham, MA, USA) and 7-amino-actinomycin staining (7AAD) (eBioscience^TM,^ Waltham, MA, USA). ANX V/7AAD double staining is a convenient way to discriminate early apoptosis from late apoptosis and necrosis. In particular, during early chondrocyte apoptosis, phosphatidylserine is translocated to the outer layer of the cell membrane, and it binds to the transmembrane protein ANX V, whose on-surface staining can be therefore used to detect apoptosis [43].

Analysis was performed in LSRII cytometer using FACS Diva6 (Becton Dickinson, Franklin Lakes, NJ, USA) and FlowJoX software (Ashland, OR, USA), version 9, for data processing. Imaging flow cytometry was performed using the ImageStream^®^X Mark II Imaging Flow Cytometer (Merck Millipore, Billerica, MA, USA) with INSPIRE software (Merck Millipore, Billerica, MA, USA), then evaluated in IDEAS software (Merck Millipore, Billerica, MA, USA) version 6.0.

### 4.7. Real-Time Quantitative PCR Analysis

Cells were homogenized using TRIzol (Thermo Scientific, Waltham, MA, USA) (500 µL for 1 million cells), and RNA extraction was performed using RNA Clean & Concentrator5-Zymoresearch. Complementary DNA was synthesized from RNA with Superscript II-Invitrogen^TM^. Real time quantitative PCR on specified gene primers (Table 1) was performed with LightCycler480^®^system (Roche Applied Science, Penzberg, Germany). Results were normalized on Gapdh as a housekeeping gene, and delta-delta CT was calculated on control samples.

### 4.8. Lysyl Oxidase Activity

LOX activity was measured in supernatants of WT and CD11b KO chondrocytes cultured in DMEM without phenol red for 48 h by use of PromoKine Lysyl Oxidase Activity Assay Kit from PromoCell (Heidelberg, Germany).

### 4.9. Collagen Content and Collagen Crosslinks

WT and CD11b KO chondrocytes were cultured in T75 flasks for 3 weeks in DMEM low glucose (10% FBS, 1× MEM NEEA, 50 µg/mL ascorbic acid). Then, cell layers were washed with PBS, scraped, and stored for collagen analysis at −20 °C. Analysis of crosslinks (dihydroxylysinonorleucine (DHLNL), hydroxylysinonorleucine (HLNL), and histidinohydroxymerodesmosine (HHMD)) was performed as described previously [44,45]. The nomenclature used in this manuscript refers to the reduced variants of crosslinks (DHLNL, HLNL, HHMD). Collagen content was measured and used for normalization of crosslink values.

### 4.10. Statistical Analysis

All data were calculated by mean ± standard deviation of triplicates or more. Graphs were made in GraphPad Prism software (San Diego, CA, USA), version 9. Variation amongst genotypes and experimental conditions were calculated using Student’s *t*-test or one-way or two-way Anova, depending on the type of data. Statistical significances were shown as follows: * *p* < 0.05, ** *p* < 0.01, *** *p* < 0.001, **** *p* < 0.0001.

## Figures and Tables

**Figure 1 ijms-24-09776-f001:**
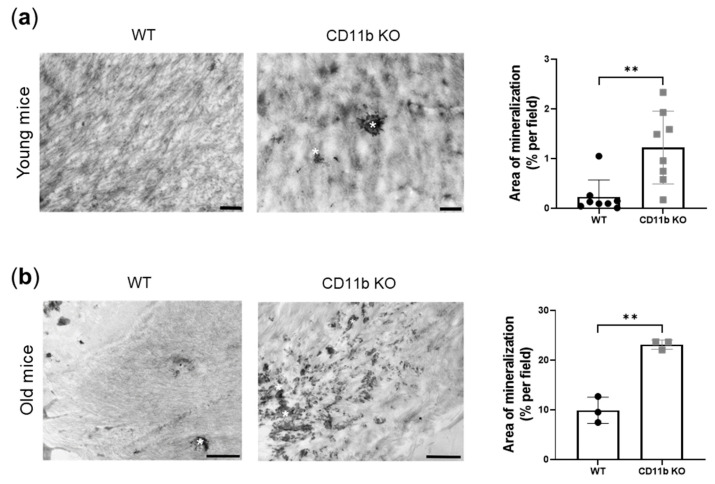
CD11b deficiency promotes articular cartilage calcification. (**a**) Representative transmission electron images of crystal deposits (asterisks) in the ECM of 13-week-old WT and CD11b KO mice and corresponding quantification (% per field) by ImageJ. (**b**) Representative transmission electron images of crystal deposits (asterisks) in the ECM of 77-week-old WT and CD11b KO mice and corresponding quantification (% per field) by ImageJ. Scale bars 1 µm. ** *p* < 0.01.

**Figure 2 ijms-24-09776-f002:**
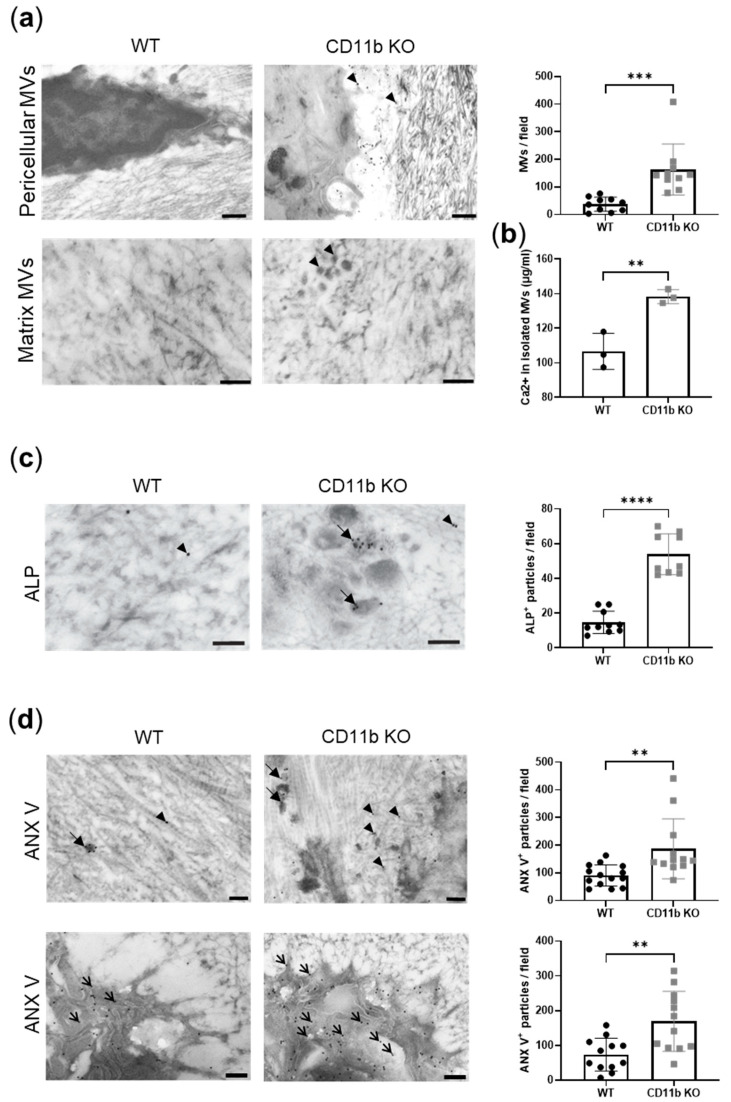
CD11b deficiency is associated with increased matrix vesicle release and calcification proteins. (**a**) Representative transmission electron images showing the presence of electron-dense extracellular vesicles (arrowheads) in the ECM of CD11b KO cartilage. Scale bars 1μm upper panels, 500 nm lower panels. Graph corresponds to MV quantification per field. (**b**) Calcium content in MVs isolated from serial ultracentrifugation of supernatants from WT and CD11b KO chondrocytes cultured 24 h in DMEM + 10% FBS. (**c**) ALP immuno-gold labeling in WT and CD11b KO cartilage and corresponding quantification. Scale bars 200 nm. Arrowheads = ECM ALP; bold arrows = MVs ALP. (**d**) ANX V immuno-gold labeling in WT and CD11b KO cartilage (extracellular in top panels; cellular in bottom panels) and corresponding quantification. Scale bars 200 nm. Arrowheads = ECM ANX V; bold arrows = MVs ANX V; thin arrows = cellular ANX V. All data are from 13-week-old mice. ** *p* < 0.01, *** *p* < 0.001, **** *p* < 0.0001.

**Figure 3 ijms-24-09776-f003:**
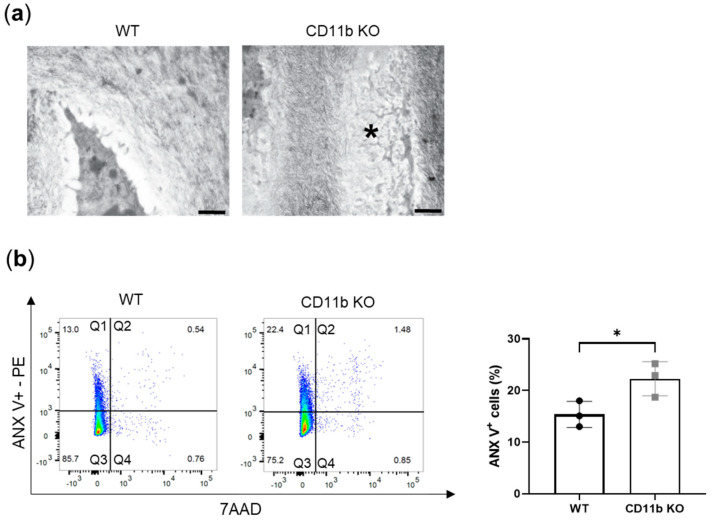
Apoptosis is increased in CD11b-deficient chondrocytes. (**a**) Representative transmission electron images of a healthy chondrocyte in WT cartilage and the remaining parts of a dead chondrocyte (asterisk) in CD11b KO cartilage. Scale bars 200 nm. Data are from 13-week-old mice. (**b**) FACS analysis for early apoptosis (ANX V-PE, *y*-axis) and for late apoptosis and necrosis (7AAD, *x*-axis). Graph corresponds to quadrant 1 (Q1) for triplicate samples. * *p* < 0.05.

**Figure 4 ijms-24-09776-f004:**
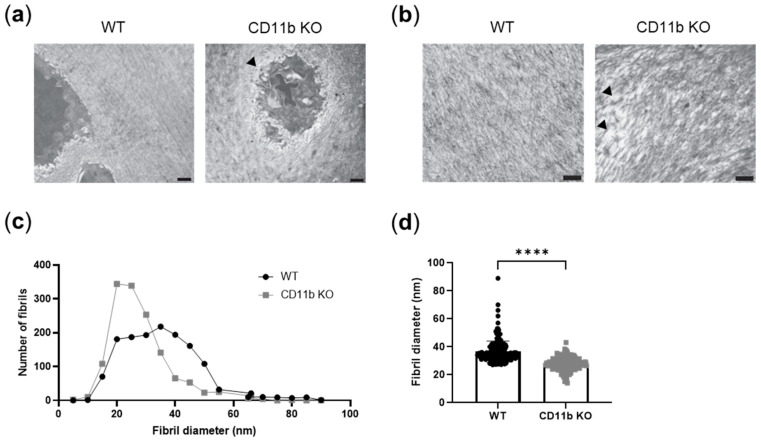
ECM organization is altered in CD11b-deficient cartilage. (**a**) Representative transmission electron images of a healthy WT chondrocyte surrounded by an intact pericellular matrix and a CD11b KO chondrocyte with alterations in the pericellular matrix (arrowhead). Scale bars 1 µm. (**b**) Representative images showing dispersed collagen fibrils (arrowheads) in CD11b KO ECM. Scale bars 1 µm. (**c**) Distribution of collagen fibrils’ number and diameter in WT and CD11b KO cartilage matrices, measured by ImageJ. (**d**) Fibril diameter in WT and CD11b KO cartilage ECM measured by ImageJ. All data are from 13-week-old mice. **** *p* < 0.0001.

**Figure 5 ijms-24-09776-f005:**
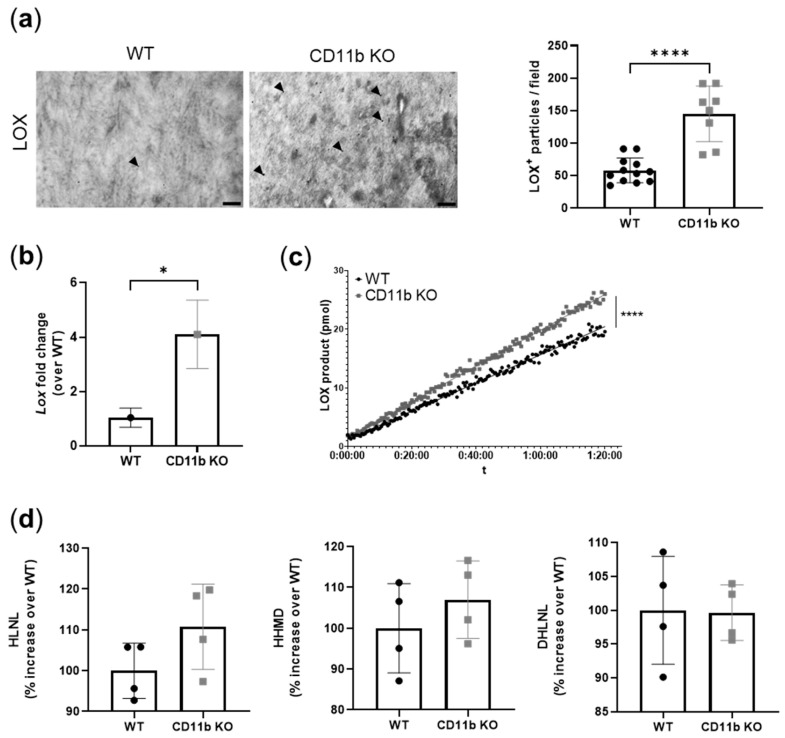
CD11b KO cartilage presents increased LOX expression and activity. (**a**) LOX immuno-gold labeling in WT and CD11b KO cartilage and corresponding quantification. Scale bars 200 nm. Arrowheads = LOX. Data are from 13-week-old mice. (**b**) qPCR analysis of *Lox* gene in WT and CD11b KO chondrocytes. Graph represents the mean ± SD of triplicates. (**c**) LOX activity, measured as LOX product over time, in the supernatant of WT and CD11b KO chondrocytes. Graph is from one experiment of two independent experiments. (**d**) HLNL, HHMD, and DHLNL crosslinks, normalized over collagen content, in WT and CD11b KO chondrocytes. WT condition was considered as reference (=100%). * *p* < 0.05, **** *p* < 0.0001.

**Figure 6 ijms-24-09776-f006:**
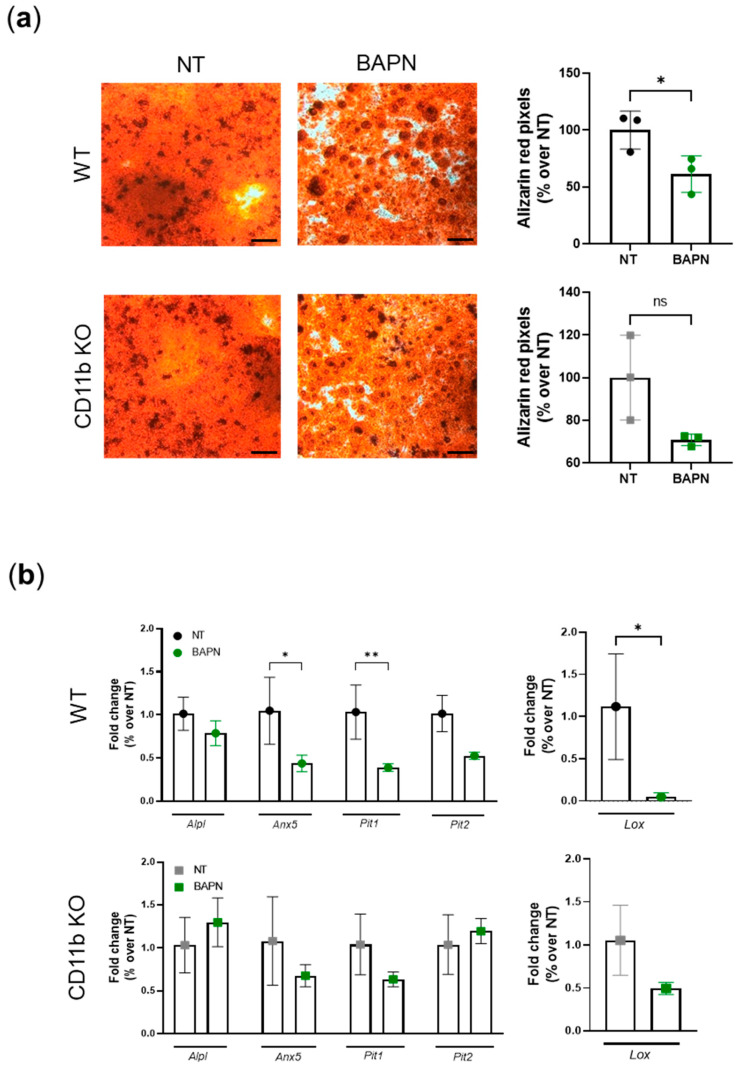
CD11b partially acts via sensing matrix crosslinks: (**a**) In step 1, we cultured WT chondrocytes in not-treated (NT) medium for 21 days in presence or absence of BAPN 500 µM to obtain normal or lowly crosslinked matrices. In step 2, after removal of chondrocytes, we cultured a new set of chondrocytes on these pre-formed matrices for 3 days in calcifying medium. Pictures depict alizarin-red-stained chondrocytes and graphs show corresponding quantification by color binarization in Adobe^®^ Photoshop^®^ (San Jose, CA, USA). Pictures represent triplicates from one experiment of two independent experiments. Scale bars 500µm. (**b**) qRT-PCR analysis of the indicated genes in murine chondrocytes cultured for 3 days in calcifying medium on pre-formed matrices. Results represent mean ± SD of triplicate samples. * *p* < 0.05, ** *p* < 0.01.

**Figure 7 ijms-24-09776-f007:**
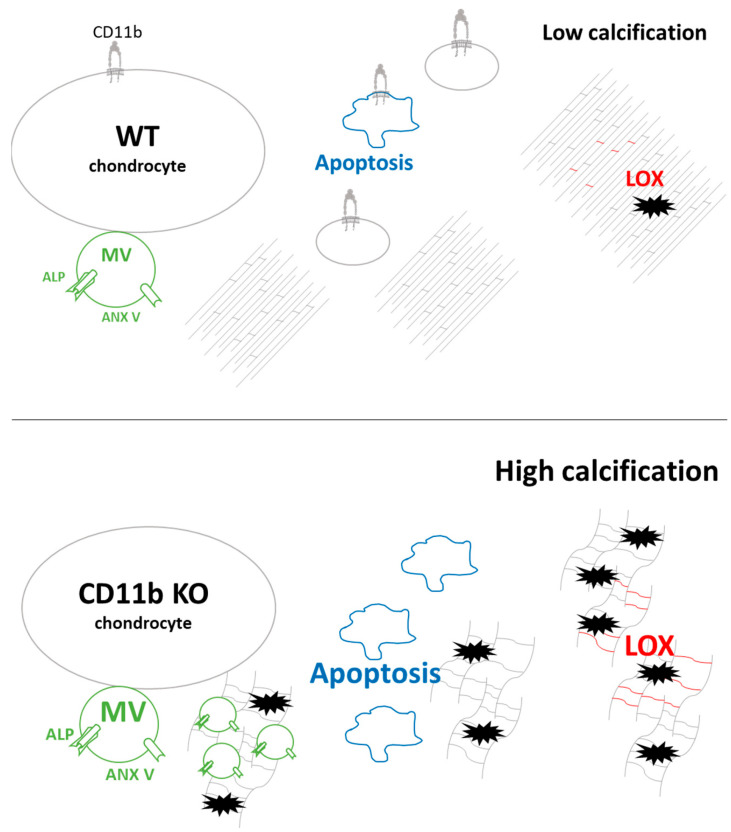
Proposed model of CD11b roles in favoring calcification of the cartilage extracellular matrix. The possible mechanisms involved are: (1) increased matrix vesicles’ (MVs) secretion, expressing ALP and ANX V (in green); (2) increased apoptosis (in blue); and (3) increased LOX activity and matrix crosslinks (in red).

**Table 1 ijms-24-09776-t001:** RT-qPCR gene primers used.

Primer Gene	Forward Primer (5′ → 3′)	Reverse Primer (3′ → 5′)
*Gapdh*	CTC ATG ACC ACA GTC CAT GC	CAC ATT GGG GGT AGG AAC AC
*Alpl*	TTG TGC CAG AGA AAG AGA GAG	GTT TCA GGG CAT TTT TCA AGG T
*Anx5*	CCT CAC GAC TCT ACG ATG CC	AGC CTG GAA CAA TGC CTG AG
*Pit1*	CTC TCC GCT GCT TTC TGG TA	AGA GGT TGA TTC CGA TTG TGC
*Pit2*	AAA CGC TAA TGG CTG GGG AA	AAC CAG GAG GCG ACA ATC TT
*Lox*	CAC TGC ACA CAC ACA GGG AT	TGT CCA AAC ACC AGG TAC GG

## Data Availability

Data are presented in this manuscript. Raw data are available upon request to the corresponding author.

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
