# Peer review of "CD11b Deficiency Favors Cartilage Calcification via Increased Matrix Vesicles, Apoptosis, and Lysyl Oxidase Activity"

_ijms, 2023, doi:10.3390/ijms24119776_

Round 1
Reviewer 1 Report
The authors have presented a very nice work on how the integrin subunit CD11b plays a role in the calcification of cartilage in mice. The work is well presented and is methodologically sound. I recommend this paper for publication, after minor revisions.
2.1 Could you use a larger font for the y axis in Figure 1a?
2.2 Could you use larger images for Figure 2?
''The dimensions of these vesicles were between 50 nm and 200 nm, which is in agreement with the size of MVs'' Could you add a reference for this?
2.3 ''By TEM 124 analysis we found much more signs of apoptotic chondrocytes in cartilage from naïve 125 CD11b KO mice than in cartilage from WT mice''. Could you please describe the morphological differences, as the image is not very clear.
2.5 Have you mechanically characterize the matrices?
4.1 Have you used the mice under an animal licence? Please give details about this.
4.2 Which concentration of secondary antibody was used? Please specify.
4.3 What does ''NT'' stand for? Did you characterize the chondrocytes?
4.4 Could you add a figure representing the Two-step experiment?
4.5 Could you explain at which time point you have collected the MVs from the supernatant? Could you please specify the centrifuge's model?
Author Response
The authors have presented a very nice work on how the integrin subunit CD11b plays a role in the calcification of cartilage in mice. The work is well presented and is methodologically sound. I recommend this paper for publication, after minor revisions.
We thank the reviewer for the time spent reading our manuscript. Here are the answers, point by point:
2.1 Could you use a larger font for the y axis in Figure 1a?
We agree with the reviewer. The font of the y axes in Figure 1 have been increased from 12 to 16.
2.2 Could you use larger images for Figure 2?
Figure 2 has been modified: Figure 2b has been placed below the quantifications of MVs in TEM of Figure 2a, thus allowing us to increase the size of the images in respect to a A4 page.
''The dimensions of these vesicles were between 50 nm and 200 nm, which is in agreement with the size of MVs'' Could you add a reference for this?
We thank the reviewer for noticing this detail, we have now added in line 98 the proper reference [8] (J.M. Carnino, H. Lee, Y. Jin, Isolation and characterization of extracellular vesicles from Broncho-alveolar lavage fluid: a review and comparison of different methods, Respir Res 20(1) (2019) 240.)
2.3 ''By TEM analysis we found much more signs of apoptotic chondrocytes in cartilage from naïve CD11b KO mice than in cartilage from WT mice''. Could you please describe the morphological differences, as the image is not very clear.
We thank the reviewer for this question. We have now added an explanation from our TEM expert (co-author Dr. Uwe Hansen) in the results section (lines 129-132): “By TEM analysis we found much more signs of apoptotic chondrocytes in cartilage from naïve CD11b KO mice than in cartilage from WT mice. Indeed, the electron micrograph in Figure 3a shows an area within the extracellular matrix with the remaining parts of a dead chondrocyte (asterisk). The area has the same shape and size of a chondrocyte but the cytoplasm with the organelles is degraded and visible as numerous amorphous structures. In addition, the degraded chromatin appears as more electron-dense structures in this area.”
We have additionally clarified in the Figure 3a legend (line 144): “Representative transmission electron images of a healthy chondrocyte in WT cartilage and the remaining parts of a dead chondrocyte (asterisk) in CD11b KO cartilage.”
2.5 Have you mechanically characterize the matrices?
We thank the reviewer for this question. In figure 6a, we did not characterize the matrices mechanically. Nevertheless, the effect of the LOX(L) inhibitor BAPN in diminishing extracellular matrix cross-links has been already well described in two studies. We therefore modified the text in the results section, (line 189-193) under “2.5 CD11b partially acts via sensing matrix cross-links”: “Briefly, we cultured WT chondrocytes in absence (NT) or presence of BAPN in order to obtain matrices with normal or decreased amount of cross-links, as described in other studies [28, 29] (Figure S1). In a second step, WT and CD11b KO chondrocytes were replated on these pre-formed matrices and let to calcify. We found that WT cells calcified significantly less on lowly cross-linked matrices (BAPN) that on normally cross-linked matrices (NT) (Figure 6a, WT, alizarin red pictures and graph).”
4.1 Have you used the mice under an animal licence? Please give details about this.
The authorization number is already reported at the end of the methods section (line 425): “Institutional Review Board Statement: Chondrocytes isolation and knees extraction from mice was done in accordance with the Swiss Federal Regulations, under the “Service de la consommation et des affaires vétérinaires du Canton de Vaud” Switzerland (No VD2711.1a).”
4.2 Which concentration of secondary antibody was used? Please specify.
In the methods section (line 327), under “4.2. Transmission electron microscopy (TEM) and immuno-gold labeling”, we modified as follows: “After washing, with the same solution, ultrathin sections were incubated with secondary antibodies (diluted 1:20) conjugated to 18 nm gold particles.”
4.3 What does ''NT'' stand for? Did you characterize the chondrocytes?
In the methods sections (line 339), under “4.3. Murine articular chondrocytes”, we corrected as follows: Chondrocytes were isolated from WT and CD11b KO mice between 4 and 7 days old, as previously described , and amplified for 7 days in not treated medium (NT = DMEM with 10 % FBS).
Additionally, in the legend of Figure 6 (line 207), we added: “(a) In step 1, we cultured WT chondrocytes in not treated (NT) medium”
As for the characterization of the chondrocytes, first we always used maximum passage 2 chondrocytes, in order to be sure that they do not undergo differentiation; second, we routinely check by qRT-PCR the proper chondrocytic commitment of these cells by expression of typical chondrocytes markers, such as Sox9 and Col2; third, we always observe cell morphology by microscopy that confirms hexagonal shape of these cells, as established in the original protocol ref [42].
4.4 Could you add a figure representing the Two-step experiment?
We added the required figure in supplementary materials, figure S1. We have added a reference to this figure in the results section (line 191) and methods section (line 347).
4.5 Could you explain at which time point you have collected the MVs from the supernatant? Could you please specify the centrifuge's model?
To answer both points of the reviewer, in the Methods section under “4.5. MVs isolation and calcium content” (lines 358-360), we rewrote: “Extracellular vesicles, of 30-150 nm diameter (corresponding to matrix vesicles, MVs) were isolated from chondrocytes supernatants by serial ultracentrifugations at 4 °C [8], using Optima™ XPN-80 (Beckman Coulter). Briefly, WT and CD11b KO chondrocytes were cultured 24h in DMEM with 10% FBS. Supernatants were collected and centrifuged at 300 x g for 10 minutes to remove dead cells and debris.”
Additionally, in the figure 2b legend (lines 119-120), we added: “Calcium content in MVs isolated from serial ultracentrifugation of supernatants from WT and CD11b KO chondrocytes cultured 24h in DMEM + 10% FBS.”
Reviewer 2 Report
The manuscript investigates the possible mechanism by which CD11b deficiency could favor cartilage calcification in naïve female mice. They showed that CD11b KO cartilage from young mice presented early calcification spots compared to WT. CD11b KO cartilage from old mice showed progression of calcification areas.
They proposed that the calcification observed in the CD11b KO cartilage mice is associated with the increased expression of lysyl oxidase (LOX). They also showed that the calcium flux in the cartilaginous cells is mediated by Annexin V.
The manuscript is interesting but some points need to be clarified.
-Line 222: the authors showed the involvement of Annexin V in the different cartilaginous cells by immunohistochemistry but did not investigate other ion channels relevant in bone and cartilaginous formations like the TRP channels including the TRPV1. The involvement of these ion channels in bone (Scala et al., 2019, 2022) and chondrocyte (Savadipour et al., 2022) cannot be excluded and need to be considered at least in the discussion section as a possible mechanism of calcium flux regulation (Scala et al., 2019, 2022).
- Line 283: The authors performed experiments in female mice, they at least explain why in the methods and in the discussion section
- Line 386: The authors used 4 mice x groups, and the sample size was not calculated x age, please explain why ??? The sample size is obligatory in the EU regulation related to animal use for scientific purposes.
-The CD11KO mice used at different ages and some were neonates are suffering phenotypes ??? Please advise that a suffering phenotype may affect the gene expression levels ....
Regulation of chondrocyte biosynthetic activity by dynamic hydrostatic pressure: the role of TRP channels.
Connect Tissue Res. 2022 Jan;63(1):69-81. doi: 10.1080/03008207.2020.1871475. Epub 2021 Jan 25.
Bisphosphonates Targeting Ion Channels and Musculoskeletal Effects. Front Pharmacol. 2022 Mar 15;13:837534. doi: 10.3389/fphar.2022.837534. eCollection 2022. Zoledronic Acid Modulation of TRPV1 Channel Currents in Osteoblast Cell Line and Native Rat and Mouse Bone Marrow-Derived Osteoblasts: Cell Proliferation and Mineralization Effect. Cancers (Basel). 2019 Feb 11;11(2):206. doi: 10.3390/cancers11020206.
minor editing revision:
line 382
Author Response
The manuscript investigates the possible mechanism by which CD11b deficiency could favor cartilage calcification in naïve female mice. They showed that CD11b KO cartilage from young mice presented early calcification spots compared to WT. CD11b KO cartilage from old mice showed progression of calcification areas.
They proposed that the calcification observed in the CD11b KO cartilage mice is associated with the increased expression of lysyl oxidase (LOX). They also showed that the calcium flux in the cartilaginous cells is mediated by Annexin V.
The manuscript is interesting but some points need to be clarified.
We thank the reviewer for the time spent reading our manuscript. Here are the answers, point by point:
-Line 222: the authors showed the involvement of Annexin V in the different cartilaginous cells by immunohistochemistry but did not investigate other ion channels relevant in bone and cartilaginous formations like the TRP channels including the TRPV1. The involvement of these ion channels in bone (Scala et al., 2019, 2022) and chondrocyte (Savadipour et al., 2022) cannot be excluded and need to be considered at least in the discussion section as a possible mechanism of calcium flux regulation (Scala et al., 2019, 2022).
We have now added a paragraph in the discussion section (lines 238-242): “Here, higher number of MVs was accompanied by increased ALP and ANX V staining in CD11b KO cartilage (Figure 7, green). However, we did not investigate whether other Ca2+ channels than ANX V were involved in chondrocyte calcification. These could encompass TRPV1 channel, shown to contribute to osteoblast mineralization [31], or TRPV4 channel involved in chondrocyte calcification [32].”
- Line 283: The authors performed experiments in female mice, they at least explain why in the methods and in the discussion section
As mentioned in the manuscript, we previously published a paper titled “CD11b Signaling Prevents Chondrocyte Mineralization and Attenuates the Severity of Osteoarthritis” 2020 Front. Cell Dev. Biol. In this, the mice used for the in vivo model of cartilage calcification were females. We therefore decided to consistently used this gender to clarify the underlying mechanisms involved. Of interest, female mice are less affected by cartilage calcification than male mice, therefore we can be sure that conclusions drawn in female mice would be true also for male mice.
- Line 386: The authors used 4 mice x groups, and the sample size was not calculated x age, please explain why ??? The sample size is obligatory in the EU regulation related to animal use for scientific purposes.
The choice of number of mice was limited by the cost and time consuming TEM experiments. Nevertheless, this sample size was validated and accepted by the ethical committee (Authorization number: VD2711.1a). Indeed, to obviate to this issue, we analysed a high number of fields per mouse for TEM, as shown by scatter plot graphs.
-The CD11KO mice used at different ages and some were neonates are suffering phenotypes ??? Please advise that a suffering phenotype may affect the gene expression levels ....
The CD11b KO mice used do not exhibit a suffering phenotype. Please refer to the below reported data sheet that was submitted together with the animal authorization (n° VD2711.1a)

Round 2
Reviewer 2 Report
the manuscript is improved
the manuscript is improved